# Molecular Mechanisms of RSV and Air Pollution Interaction: A Scoping Review

**DOI:** 10.3390/ijms232012704

**Published:** 2022-10-21

**Authors:** August Wrotek, Teresa Jackowska

**Affiliations:** 1Department of Pediatrics, Centre of Postgraduate Medical Education, Marymoncka 99/103, 01-813 Warsaw, Poland; 2Department of Pediatrics, Bielanski Hospital, Cegłowska 80, 01-809 Warsaw, Poland

**Keywords:** respiratory syncytial virus, air pollution, particulate matter, PM_2.5_, pathomechanism, environmental pollutants, bronchial hyperreactivity

## Abstract

RSV is one of the major infectious agents in paediatrics, and its relationship with air pollution is frequently observed. However, the molecular basis of this interaction is sparsely reported. We sought to systematically review the existing body of literature and identify the knowledge gaps to answer the question: which molecular mechanisms are implied in the air pollutants–RSV interaction? Online databases were searched for original studies published before August 2022 focusing on molecular mechanisms of the interaction. The studies were charted and a narrative synthesis was based upon three expected directions of influence: a facilitated viral entry, an altered viral replication, and an inappropriate host reaction. We identified 25 studies published between 1993 and 2020 (without a noticeable increase in the number of studies) that were performed in human (n = 12), animal (n = 10) or mixed (n = 3) models, and analysed mainly cigarette smoke (n = 11), particulate matter (n = 4), nanoparticles (n = 3), and carbon black (n = 2). The data on a damage to the epithelial barrier supports the hypothesis of facilitated viral entry; one study also reported accelerated viral entry upon an RSV conjugation to particulate matter. Air pollution may result in the predominance of necrosis over apoptosis, and, as an effect, an increased viral load was reported. Similarly, air pollution mitigates epithelium function with decreased IFN-γ and Clara cell secretory protein levels and decreased immune response. Immune response might also be diminished due to a decreased viral uptake by alveolar macrophages and a suppressed function of dendritic cells. On the other hand, an exuberant inflammatory response might be triggered by air pollution and provoke airway hyperresponsiveness (AHR), prolonged lung infiltration, and tissue remodeling, including a formation of emphysema. AHR is mediated mostly by increased IFN-γ and RANTES concentrations, while the risk of emphysema was related to the activation of the IL-17 → MCP-1 → MMP-9 → MMP-12 axis. There is a significant lack of evidence on the molecular basics of the RSV–air pollution interaction, which may present a serious problem with regards to future actions against air pollution effects. The major knowledge gaps concern air pollutants (mostly the influence of cigarette smoke was investigated), the mechanisms facilitating an acute infection or a worse disease course (since it might help plan short-term, especially non-pharmacological, interventions), and the mechanisms of an inadequate response to the infection (which may lead to a prolonged course of an acute infection and long-term sequelae). Thus far, the evidence is insufficient regarding the broadness and complexity of the interaction, and future studies should focus on common mechanisms stimulated by various air pollutants and a comparison of influence of the different contaminants at various concentrations.

## 1. Introduction

Air pollution is one of the major public health concerns due to a well-established and still growing body of evidence on its detrimental effects on human health; according to the World Health Organization (WHO), air pollution is estimated to contributs to approximately 7 million premature deaths globally [1,2]. Air pollutants may be classified depending on their 1. Formation—primary (emitted directly) versus secondary (formed as a result of reactions with other pollutants or gases), 2. Origin—indoor versus outdoor, or 3. physical characteristics—gaseous (nitrogen oxides, NOx, sulfur dioxide, SO_2_, ozone, O_3_, carbon monoxide, CO, specific volatile organic compounds, SVOC (e.g., benzene) versus particulate (particulate matter smaller than or equal to 10 µm in diameter, PM_10_ or coarse PM; particulate matter between 0.1 and 2.5 µm in diameter, PM_2.5_ or fine PM; particulate matter smaller than or equal to 0.1 µm in diameter, ultrafine PM) [3]. The studies on the influence of air pollution on human health have mainly revealed its relationship with cardiovascular and respiratory diseases; however, outdoor air pollution and particulate matter are also listed by the WHO agency and the International Agency for Research on Cancer (IARC) as carcinogenic hazards to humans [4,5]. Thus, the WHO announced that the six most significant air pollutants include particulate matter smaller than or equal to 2.5 µm in diameter (PM_2.5_), particulate matter smaller than or equal to 10 µm in diameter (PM_10_), ozone (O₃), nitrogen dioxide (NO₂), sulfur dioxide (SO₂), and carbon monoxide (CO), and the updates on the highest tolerable levels of those pollutants are published in WHO Global Air Quality Guidelines (AQGs) on a regular basis [1]. 

Respiratory syncytial virus (RSV) is one of the most significant etiological agents of respiratory tract infections (RTI), both due to its burden, as well as clinical severity; the global estimates report RSV as the cause of 33 million lower RTI (LRTI) episodes and 3.6 million of hospitalizations per year, and those numbers are restricted to children under 5 years of age only [6]. Two percent of deaths in children under 5 years of age, and 3.6% in children aged 1–6 months are related to RSV [6]. Moreover, RSV, although initially considered to be a virus of infancy only, turned out to be one of the major infective factors in adults, especially in the elderly [7,8,9]; despite a limited access to viral testing and often unspecific clinical picture of RSV RTI, the awareness of the significance of RSV is growing [10]. The high frequency and severity of the RSV disease translate into a huge socioeconomic impact and utilization of healthcare resources seen in paediatric and adult population [7,11,12,13], urging a development of monoclonal antibodies and vaccines targeting infants, pregnant women and older adults [14]. A relationship between lower air quality and RSV infections was observed mainly in hospital-based and some community-based studies [15,16,17,18,19,20,21], with particular interest in PM_2.5_ [17,18,21], PM_10_ [15,21], nitric dioxide (NO_2_) [21], ozone [22], carbon monoxide (CO) [20], and benzene [19]. Exposure to another pollutant, cigarette smoke, is a well-described risk factor for a more severe RSV infection in children, an increased RSV LRTI prevalence [23,24,25], and chronic obstructive pulmonary disease (COPD) [26]; in the latter case, the RSV can be persistently present, escaping an immune response [27]. 

Air pollutants induce a complex interaction via various pathogenetic pathways in the respiratory tract, including an increased production of reactive oxygen species (ROS), an activation of transcription factors, or the production of cytokines and/or chemokines; as a result, local homeostasis is disturbed, lung immunity is reduced, inflammatory response is exacerbated, and hyperreactivity is increased [28]. While some of the mechanisms are common and expected to be repeatedly present in a response to various etiological factors, the other ones might be pathogen-specific [28]. Nonetheless, an in-depth knowledge on the molecular bases of the air pollutants–RSV interaction is indispensable for planning any wide-spread action counteracting the effects of air pollution. Moreover, when a specific mechanism is identified, targeted actions, including pharmacological treatment, might be considered. The rationale for this systematic review is based upon a growing number of evidence on the influence of air pollutants on the RSV morbidity, two issues with a huge socioeconomic impact. In this study, we sought to systematically review the existing body of literature and identify the knowledge gaps in the context of the mechanisms of interaction between the RSV and air pollutants on the molecular level. For this purpose, we formulated the following research question: which molecular mechanisms are implied in the interaction between RSV and the particular air pollutants?

## 2. Materials and Methods

### 2.1. Study Protocol

The study protocol was prepared with the use of the Preferred Reporting Items for Systematic reviews and Meta-Analyses extension for Scoping Reviews (PRISMA-ScR) and is available on request from the corresponding author.

#### 2.1.1. Conceptual Framework

Two overlapping groups of variables were considered in the study: the air pollution and the molecular mechanisms. The concept of the air pollution was defined as broadly as possible in pursuance of a better understanding of the mechanisms that are involved in the pollutant–virus interaction. We assumed that at least some of the pathological pathways are initiated/enhanced/blocked by various pollutants with a similar final result, although due to the major differences between the pollutants, some of the mechanisms cannot be adapted to another pollutant–RSV interaction and require a confirmation in further separate studies. Furthermore, some of the mechanisms might give a false impression of being common, since, for example, diesel exhaust contains different contaminants (carbon monoxide, sulfur dioxide, nitrogen oxides, etc.) that might and should be analysed separately [29]; thus, in the result presentation, we always underline the investigated pollutant with regards to a particular mechanism. The molecular basis might include a broad spectrum of actions, mediated by various receptors, cytokines, gene expression, etc. The search contained a vast number of terms and keywords that might be related to molecular mechanisms (see the search strategy in the Appendix A). Although the effects of air pollution on human beings are complex and may overlap, three major directions of unfavourable air pollution impact might be expected (Figure 1): 

a facilitated viral entry via an enhanced viral adhesion (as a result of expression/increased expression of the potential receptor/adhesion molecule on the cell surface), or facilitated entry due to the presence of a transporting canal or damaged natural protective barriers;an altered viral load (presumably an increased/prolonged viral replication) via either the virus gaining an unusual virulence or ability (per se) to replicate or a decreased antiviral defense of the host, or via a facilitated/increased viral release;an inappropriate host reaction, including but not limited to a prolonged or increased inflammatory reaction, hyperreactivity, or histopathological changes; may result in a more severe disease course, more damage to the host, and the presence of long-term sequelae.

#### 2.1.2. Review Approach

In order ensure the highest possible quality of the result reporting, we used a systematic review approach to perform a scoping review in accordance with the PRISMA extension for scoping reviews (Appendix A) [30]. The scoping review format was chosen since the main purpose of the study was to identify the knowledge gaps and scope of a body of the literature [31]. 

#### 2.1.3. Search Strategy

To identify the potentially relevant papers published before 1 August 2022, the following three bibliographic databases were searched: PubMed, Scopus, and Cochrane. The last search was executed on 16 August 2022. The search strategy was based on 3 major categories: A. Etiology, B. Air pollution, and C. Molecular mechanism of the interaction, which were then combined with the Boolean operator “AND”. The primary search was performed in PubMed, with the use of MeSH terms and keywords with synonyms regarding the area of interest; then, the search was converted in order to search the other databases. The exact search strategy is presented in the Appendix A. In order to retrieve as much literature in the area of interest as possible, we undertook an analysis of the reference lists of the included studies as well as review articles which were excluded from the analysis but might have potentially revealed additional studies that would meet search criteria; however, our strategy did not include a gray literature search and was limited to peer-reviewed and published literature only. The selection of sources of evidence: the databases were searched independently by the authors, and a two-step process of 1. title and abstract review, and 2. a full text analysis was performed. To ensure consistency between the authors, after screening of the most recent 50 publications (found in PubMed search), a comparison of the screening results was performed and discussed; after this calibration, the proper screening was executed. In dubious cases, the results of the review were compared and an agreement was reached.

#### 2.1.4. Eligibility Criteria

We included only original papers analysing a combined interaction between air pollution and RSV, regardless of the studied model (i.e., cellular, animal, or human). We excluded the studies that investigated air pollution or RSV influence alone, or did not analyse the mechanism underlying the interaction or focused on prenatal exposure only. On the other hand, there were no restrictions in terms of the date of publication, language, study location, study setting or design, the sequence of exposure to air pollution and infection, including a co-exposure, nor was the study funding source an excluding factor. A detailed list of inclusion and exclusion criteria is shown in Table 1.

#### 2.1.5. Extraction Methods

An extraction grid for data-charting was created cooperatively by the reviewers, and general information on the study (first author, year of publication, title, journal, geographic region by country), data regarding the subject and the interaction (i.e., investigated model, air pollutant, viral exposure, including additional data on, for example, repeated exposure and the analysed mechanisms), as well as the most significant results was collected. No simplifications were made, especially on the studied air pollutant and model used. The extraction was performed independently by the authors. The extraction grid is presented in the results section in a shortened version as a table (Table 2 in the results section regarding the included studies and as Appendix A with the characteristics of the studies). 

#### 2.1.6. Critical Appraisal

In order to ensure a thorough reporting, a methodological quality assessment of the included studies was undertaken, although it is not required in a scoping review. A choice of appropriate tool is a complex issue; many instruments have been developed, including scales, checklists, or items [56]. Additionally, in the case of this scoping review, different types of the studies were included. We chose to use one quality assessment tool for two reasons: 1. an inclusion of mixed human/animal models, 2. lack of information on the domains, which are significant parts of risk of bias assessment in animal studies, such as allocation sequence, animal housing, and blinding of caregivers/investigators/assessors [57,58]. The latter seems obvious, but due to limited space in the published papers, the majority of the studies do not report it, and refer to the guidelines that were followed. Thus, in the case of animal (or mixed) models, we added a reference list of the animal studies guidelines if they were literally mentioned in the studies. 

The Joanna Briggs Institute (JBI), except for publishing a guidance for the conduct of scoping reviews, also offers a number of widely-used critical appraisal tools, and an inter alia Checklist for Quasi-Experimental Studies (non-randomized experimental studies), which was chosen due to protocols followed by the majority of the studies [59]. The domains assessed by this tool are presented in the Appendix A. The assessment was conducted independently by the authors, and any disagreement was discussed and a consensus was reached. The results are presented in the table, and were interpreted separately for each study; for this purpose, we used a comprehensive approach and did not use a summary score in order to omit a risk of assigning weights to particular domains, especially taking into account that some points might have not been reported but were followed in the study and the contact with the authors might be hampered due to the period of time passed since the publication. If a significant risk of bias was suspected, it was assumed to be reported together with the study results presentation. 

#### 2.1.7. Synthesis of the Results

The included studies are gathered in a summarizing table, which includes the details of study population, the type of the air pollutant exposure, the analysed effects, and the major results. With a view to facilitate and synthesize the concepts, we chose a narrative synthesis divided into the three main directions of air pollution–RSV interactions mentioned above, which are expected to be crucial for the disease course in patients: a facilitated viral entry, an altered viral replication, and an inappropriate host reaction. Since the investigated pathways might interfere or have common effects, corresponding diagrams are presented. 

## 3. Results

### 3.1. Search Results

The database search yielded 463 papers. After 21 duplicates were removed, 442 studies were screened, and based on the title and abstract, 415 were excluded. As a result, 27 full-text articles were assessed for the eligibility, and five publications were removed: four did not analyse the RSV–air pollutant interaction directly, and one did not investigate the molecular mechanism nor the interaction. The search of the reference lists revealed three additional studies that were then included in the analysis; finally, 25 studies met the inclusion criteria. The PRISMA flow diagram [60] is shown in Figure 2. 

### 3.2. Study Characteristics

In total, 25 papers (Table 2) published between 1993 and 2020 were included; with four exceptions, the remaining studies were published in the 21st century, including 40% (n = 10) within the last decade and 64% (n = 16) within the last 15 years; the number of the studies did not show an increasing tendency, rather a stable accumulation of the data on the topic (Figure 3).

Fifteen studies were performed in human models (n = 12) or mixed human and animal models (i.e., the studies investigated the same interaction in both a human and an animal model), while the rest (n = 10) were performed in animal models (mainly murine models, n = 8, single rat model, and guinea pig model; similarly, when both models were used, the murine model was chosen as animal model). With regards to air pollutants, the vast majority of the studies (n = 11) investigated the influence of cigarette smoke or its derivatives four studies analysed the influence of PM_10_, three focused on TiO_2_-NP, 2 on carbon black, while the remaining pollutants were analysed in single studies only (Figure 4). The overwhelming majority of the studies was conducted in the USA (n = 19), followed by Canada (n = 3), the United Kingdom (n = 2), and Japan (n = 1).

The funding sources of the included studies are shown in the Appendix A. Due to the specificity of the studies, requiring specialized materials/devices, we also cited personal acknowledgements if they were mentioned in the funding section or were any type of land/lease or donation. The funding sources were mentioned in 22 out of 25 studies, and 3 reported no external funding. In general, the funding sources seem to have no impact on study performance, results, or reporting. 

The quality assessment of the included studies was positive and raised no concerns on high quality of the studies, and the majority of concerns regarded domain number 5— multiple measurements of the outcome (Table 3), which might be of huge importance in some investigations, while for the other play a less significant role; for example, results of histological studies or gene expression might differ a lot, while the use of commercially available kits for cytokine measurements is related with a lower risk of bias. In general, included studies raised no concerns on the increased risk of bias, which might lead to false conclusions.

## 4. Molecular Mechanisms

### 4.1. A Facilitated Viral Entry

#### 4.1.1. Epithelial Barrier

Epithelium forms a first-line barrier in the airways that is crucial for antiviral defense, and a damage to the airway epithelium is characteristic for an RSV infection [52,61]. The RSV has been previously shown to cause a disintegration of the epithelium and its increased permeability (so called “leaky epithelium”), mainly due to a disruption of the apical junctional complexes (AJC), which are intercellular complexes [61]. Except for a diminished integrity, the RSV also causes a remodeling of the actin cytoskeletal to which AJC is linked [52,61]. The RSV-induced epithelial disintegration might be enhanced by an exposure to nanoparticles and titanium-dioxide nanoparticles (TiO_2_-NP), which was shown in a number of studies to be a suitable model for the studies on pulmonary effects of environmental nanoparticles (see Figure 5) [36,52,62,63,64]. Both in vitro (immortalized human bronchial epithelial cells) and in vivo (murine) models of TiO_2_-NP effects, investigated by Smallcombe, confirmed the hypothesis of an augmented barrier dysfunction [52]. The barrier dysfunction comprised an amplified AJC disruption and led to an increased viral replication [52]. The molecular mechanism of the epithelium integrity disruption by TiO_2_-NP was shown to be related to an oxidative stress and a generation of reactive oxygen species (ROS) and a pretreatment with an antioxidant (N-acetylcysteine) not only attenuated the AJC disruption, but was also able to reverse the enhanced RSV infection [52]. 

#### 4.1.2. A Facilitated Viral Entry

A facilitated viral entry was reported in an in vitro mimic of PM_10_ containing the RSV that were deposited onto the airway epithelial cells [37]. As a result, an accelerated viral entry was observed upon the RSV conjugation to PM, and the mechanism involved an endocytic pathway [37]. Moreover, the RSV survival was increased when it was associated with PM_10_ in the model [37]. No studies proving enhanced viral receptors expression on the respiratory tract cells were found.

### 4.2. An Altered Viral Load

(A)An Increased Viral Load

An increased viral load, increased RSV gene and/or protein expression, or its replication outside the epidemic season was reported after exposure to various air pollutants, such as cigarette smoke, particulate matter, nanoparticles, or nitric oxide [29,36,40,43,45,49,50]. The studies on the mechanisms underlying an increase in infections report different pathways, including enhanced necrosis and decreased apoptosis, a mitigated antiviral defense via altered functions of the epithelium, alveolar macrophages or dendritic cells, or a decreased inflammatory response [29,36,40,43,45,49,50].

#### 4.2.1. Autophagy, Decreased Apoptosis, and Enhanced Necrosis

Autophagy is crucial for the immune response, since the structures of damaged cells might be cut into recyclable amino and fatty acids, which might be furtherly reused by host cells [65]. However, some viruses also present the ability to make use of the fatty/amino acids released from the damaged cells for the purposes of their own replication [66,67,68]. A selective autophagy of damaged cells is partly regulated by the nerve growth factor (NGF)/ tropomyosin receptor kinase A (TrkA) axis with NGF playing a cytoprotective role [36,69,70,71]. The RSV causes alterations in the expression of the NGF and its receptors, upregulating the NGF and TrkA and downregulating the p75(NTR), and thus protecting against virus-induced apoptosis; the mechanism was reported in distal but not in the proximal airway epithelium. When endogenous NGF diminished, bronchial epithelial cell survival was decreased [72]. Chakraborty reported an enhanced RSV infectivity in human bronchial epithelial cells exposed to titanium-dioxide nanoparticles (TiO_2_-NP) [36]. A preexposure to the TiO_2_-NP prior to an RSV infection upregulates the NGF/TrkA axis, inducing autophagy, which promotes cell necrosis and, as a result, a viral replication [36]. A co-exposure to the RSV and TiO_2_-NP results in an increased necrosis at the expense of a reduced apoptosis [36]. To the contrary, an increased apoptosis and, in consequence, a lowered viral load was reached with an experimental use of wortmannin, which is a pharmacological inhibitor of the early autophagosomal gene beclin-1 [36]. 

Similarly to nanoparticles, a preexposure to cigarette smoke extract also resulted in a higher viral load in human tracheobronchial epithelial (hTBE) cells due to a smaller degree of apoptosis [40]. In the study by Groskreutz, ELISA and TUNEL detection of apoptosis were used and revealed a decreased apoptosis as a result of inhibited caspases activation following an RSV infection [40]. Although the apoptosis in the airway epithelium was decreased, some of the cells were dying, probably due to necrosis, which promotes viral replication [40,73]. A decrease in apoptosis, on the other hand, can be reduced by pretreatment with N-acetylcysteine and aldehyde dehydrogenase, a fact that strongly suggests it is primarily mediated by reactive aldehydes, similar to the aforementioned reactive oxygen species (ROS) or acrolein [40]. 

#### 4.2.2. Decreased Antiviral Defense

##### Epithelium

Activation of the epithelium mediated by interferons (IFN) is one of the baseline antiviral defense mechanisms. IFN type I (IFN α and β) and IFN type III (IFN-Γ) act through different receptors, but its physiological activity overlaps and executes antiviral responses [74]. IFN-γ is produced mostly by T cells and NK (natural killer) cells and binds to the receptor, causing an activation of JAK-STAT (Janus kinase-signal transducer and activator of transcription) cascade; as a result, phosphorylated and dimerized Stat1 translocates to the nucleus, where it binds to the specific IFN-γ activated sequence in various genes that are being induced in order to promote antiviral activity (recruitment of immune cells, antigen presentation, cell proliferation, or apoptosis) [49,75,76]. A lack of or decreased immune response might result in enhanced viral infection. While a pretreatment with IFN-γ prior to an RSV infection resulted in a decreased viral gene mRNA expression in primary human tracheobronchial epithelial cells under the influence of cigarette smoke extract (CSE), the decrease, however, was inhibited by CSE; the same effect was observed for RSV proteins expression [49]. CSE inhibits the aforementioned IFN-γ-dependent gene expression in epithelial cells via inhibition of the signal transducer and activator of transcription 1 (Stat1) phosphorylation [49]. Of interest, antiviral effects of IFN-γ blocked by CSE might be restored by gluthathione supplementation with the N-acetylcysteine or glutathione monoethyl ester (GSH-MEE) [49]. 

Similarly, reduced levels of IFN-γ, alongside reduced IL-12, were observed in side-stream cigarette smoke (SS)-exposed mice challenged (twice) with RSV, and in accordance with the previous study, it also led to a higher RSV gene expression [50]. Unexpectedly, Phaybouth et al. noted a similar level of decrease in Clara cell secretory protein levels in the lungs of both SS-exposed and air-exposed mice after RSV reinfection; the authors speculate that a primary infection in the neonatal period may influence an immune response in the case of a reinfection [50].

Non-ciliated airway epithelial (Clara) cells in humans are mostly located in the bronchiolar epithelium and present the ability to secret the Clara cell secretory protein (CCSP, or CC-10, or CC-16) [29,77]. CCSP is an abundant immunomodulatory protein and its deficiency results in increased viral persistence, lung inflammation, airway reactivity, and mucus production elicited by an RSV infection (the results can be reversed by restoring CCSP [29,78]. Under the influence of diesel engine emissions (DEE), the number of Clara cells (which produce CCSP) was diminished, CCSP production was attenuated, and the decrease was more accentuated after a high-level of DEE exposure [29]. Clara cells also produce other immune defense cells, for example, SP-A (surfactant protein A, a member of the collectin family), which acts as an opsonin for bacteria and viruses, and DEE exposure decreased SP-A staining in epithelial and alveolar type II cells in a dose-independent manner [29]. As a result, a significantly higher RSV gene expression was observed in mice exposed to diesel engine emissions (DEE), and the exact influence of various pollutants needs to be established, since DEE contained a number of pollutants (PM, NOx, CO, and SO_2_) [29]. 

##### Alveolar Macrophages (AM)

Alveolar macrophages play a crucial role in the clearance of alveolar space; AM clear the airways from both infectious agents (bacteria and viruses) and air pollutants, especially particulate matter [79]. Human AM, but also AM in mice or guinea pigs are permissive to an RSV infection, and its permissiveness is inversely related to the maturation of AM, i.e., the more mature the AM are, the lower the degree of the RSV application [80,81,82]. The role of the alveolar macrophages (AM) in RSV uptake was demonstrated in guinea pig alveolar macrophages, where RSV yield, defined as the amount of viral replication/RSV-immunopositive cell, was decreased in AM exposed to PM_10_ [45]. A less effective elimination of the RSV due to a decreased viral uptake by AM (a decrease reaching up to 50%) was reported by Becker and Soukup in a study investigating the human bronchial epithelial cell line and human AM from bronchoalveolar lavage, BAL (obtained from volunteers); the study also reported changes in chemokine/cytokine secretion [33]. Upon an RSV infection, AM secrete IL-8, MIP-1α, MIP-1β, and MCP-1, while RANTES derives solely from the RSV-infected bronchial epithelial cells (and is decreased in the presence of AM) [33]. PM_10_ alone stimulates the release of granulocyte chemoattractant IL-8, and MIP-1α, but not MIP-1β, MCP-1 or RANTES [33]. A co-exposure to PM_10_ and RSV inhibits MCP-1 production, and does not exhibit an additive effect on MIP-1α and IL-8 levels, which can be interpreted as a decreased production of these chemokines as well, and does not affect the RANTES production (except for the decrease in the presence of AM) [33]. The role of the chemokines must be interpreted with caution here; while MCP-1 plays a significant role in monocyte chemotaxis and differentiation of T lymphocytes, its increased expression might be related with Th2-dependent airway hyperreactivity [83]. A strong emphasis needs to be put here on the differences between the diminished immune response resulting in an attenuated virus elimination by the macrophages, and an excessive immune response leading to airway hyperreactivity, since in part, they are mediated by common mechanisms.

##### Dendritic Cells

Plasmacytoid dendritic cells (pDC) play an important role in antiviral protection; a viral stimulation of the Toll-like receptor (TLR) agonists initiate a secretion of type I IFN by pDC [84,85,86]. The Type I interferon system integrates early antiviral and immunostimulatory activities [35]. RSV products (viral RNA or virus intermediate RNA; vRNA or iRNA, respectively) may enter the endosomes of pDC (which contain TLR) via the process of autophagy; then, viral nucleic acids activate TLR7, thus recruiting MyD88, which causes phosphorylation of IRF7 [35]. IRF7 that is not phosphorylated and cannot be translocated into the nucleus and cannot initiate the transcription of the type I IFN genes. Cigarette smoke prevents the TLR7 activation by viral nucleic acids, it may also alter the activation of TLR7 upon trafficking into late endosomes/lysosomes; in turn, the nuclear factor (NF)- kB is activated, thus preventing the production of inflammatory cytokines (IL-1β, for example) [35]. Cigarette smoke inhibits TLR-7 and -9 expression and stimulation by specific TLR agonists in pDC [35]. Finally, cigarette smoke decreases IFN-α production in pDC in response to RSV [35]. In addition, an RSV-induced release of IL-1β, IL-10, and CXCL10 is decreased, without changes in other cytokines and chemokines (like IL-6, TNF-α, CCL2, MIP-1α= CCL3=, RANTES= CCL5 and CXCL8) [35]. A study by Castro shows a severe suppression of crucial pDC functions after cigarette smoke exposure, with an inhibition of the production of IFN-α, IL-10, IL1β, and CXCL10 and a downregulation of TLR7 [35]. Of note, another study focusing on the PM_10_ exposure found no statistically significant influence on the secretion of IL-1β by human airway epithelial cells challenged with RSV [42].

In another series of experiments, an influence of nitric oxide on human monocyte-derived dendritic cells (MoDCs) was shown [43]. MoDCs were exposed to the RSV during the epidemic season; then, the cultures were maintained and fresh MoDCs added monthly, and outside of the RSV epidemic season, they were exposed to nitric oxide (NO), NO donors, and NO inhibitors [43]. The exposure to any of the above agents resulted in an induction of the RSV replication, showing that the virus may remain dormant and be activated by exogenous NO [43]. The authors postulate that NO might be responsible for RSV seasonality; when higher NO levels are present in the environment, an RSV replication is triggered, similarly, patients exposed to cigarette smoke, for example, may experience an RSV infection outside the season [43]. Endogenously produced NO is expected to show antiviral activity decreasing the RSV replication, and a decreased release of nitric oxide (NO) from human monocytes (deriving from blood donors) was driven mainly by cigarette smoke; however, a higher percentage of extreme NO decrease was noticed in the presence of an RSV + CSE co-exposure, compared to each stimulus alone [53,87,88,89]. Exogenous NO (contained in cigarette smoke, for example) might decrease intracellular NO production and facilitate an infection [43]. Of note, persistently infected MoDCs exhibited an increased cell survival, suggesting that the RSV persistence involves the inhibition of apoptosis [43].

(B)Viral load fluctuations

The exposure to air pollutants may also result in lowered virus copies, but it may have a temporary character [46,48]. A lower viral load was observed in cigarette smoke- exposed compared to filtered air-exposed mice on day 4 post RSV infection; however, it was much higher on day 14. The authors discuss the possibility of type I IFN response induced by cigarette smoke which initially suppresses the replication, but due to a waning response, a delayed clearance might be expected [48].

Similarly, RSV titers were lower on days 2–4 in mice exposed to carbon black in the research by Lambert [46], but by day 7, an exacerbation of infection was observed, including an increased expression of the mRNA of proinflammatory cytokines, and protein levels of TNF-α and IL-13 in the lungs [46]. Initially, the tumor necrosis factor-α (TNF-α) levels were decreased on days 1–2, together with decreased IFN-γ mRNA, IFN-γ-inducible protein (IP-10), and lymphotactin [46]. 

While the IFN-γ and TNF-α show an antiviral activity, IP-10 deriving from macrophages mediates a Th1 immunity, and activates and chemoattracts neutrophils which interact with CXCR3 (IP-10R) on Th1 cells [90,91]. During an acute phase of an RSV infection, the number of CXCR3-positive cells is significantly decreased, while its ligand levels, IP-10, are elevated [92]. After preexposure to carbon black, a decrease in IP-10 was observed, suggesting that a preference towards a Th2 allergic immune response in place of a Th1 antimicrobial defense might be expected and facilitate a further inflammation in response to the RSV infection [46]. 

(C)Virus release

A potential reason for a worsening viral infection is an enhanced virus spread; nevertheless, we found no proof on the facilitated release of infectious RSV molecules. An experiment on human alveolar macrophages (AM) exposed to ozone and infected with the RSV showed no differences in the amount of infectious RSV released on day 2 and 4, neither was a percentage of the infected AM altered by the pollutant [54]. Nitrogen dioxide study revealed no differences in the release of the infectious virus on day 2 by lower doses of NO_2_, and, intriguingly, it was decreased in cells exposed to higher doses of NO_2_ [32].

### 4.3. An Inappropriate Host Reaction

While a diminished immune response may result in a facilitated viral replication, impaired or delayed virus elimination, an exuberant inflammatory response provokes airway hyperresponsiveness (AHR), prolonged lung infiltration and/or tissue remodelling. 

#### 4.3.1. Inflammation

In the aforementioned in vitro model by Cruz-Sanchez and colleagues, an increased secretion of IL-6 and IL-8 was observed when the RSV was harboured by PM_10_ (see Figure 6) [37]. In vivo studies also reveal an exacerbated inflammatory response; a murine model of RSV pneumonia disclosed an augmented inflammation in animals that were previously exposed to TiO_2_-NP [41]. The levels of IFN-γ and RANTES (CCL5) were used as markers of the pneumonia severity in the investigation, and their values were increased in bronchoalveolar lavage fluids (BALF) from the mice, alongside with increased IL-10, whereas viral titers were not affected [41]. 

Please note: the figure has only a demonstrative character and is based on the literature found for the purposes of this review. The effects of the air pollutants might vary with regards to the presence/absence of the mechanism, its extent, and its effects. The mechanisms might be influenced by other, not verified or not shown pathways. The mechanisms might be dose-dependent, time-dependent, exposition sequence-dependent, and model-dependent (i.e., differences between the models used in the studies and human beings might be seen). The figure is simplified, and, for example, cigarette smoke is a common name for possibly different exposures (the studies used, inter alia, cigarette smoke extract, cigarette smoke condensate, or non-specified cigarette smoke exposure for different periods of time); for details, see the text and/or refer to the original articles. In addition, contradictory effects of the same air pollutant might be reported, depending on the study model.

Abbreviations: DEE—diesel engine emissions, PM_10_—particulate matter smaller than or equal to 10 µm in diameter, TiO_2_-NP—titanium-dioxide nanoparticles; CXCL (1,9)—chemokine (C-X-C motif) ligand (1, 9), GM-CSF—Granulocyte-macrophage colony-stimulating factor, IL- (1α, 6, 8, 10, 13, 17)—Interleukin- (1α, 6, 8, 10, 13, 17), IFN-γ—interferon γ, LIF—leukemia inhibitory factor, MCP-1—monocyte chemoattractant protein-1 (=CCL2), MIP-1α—macrophage inflammatory protein-1 α (=CCL3), MIF—macrophage migration inhibitory factor, MMP-(2, 8, 9, 12, 13, 16)—matrix metalloproteinase-(2, 8, 9, 12, 13, 16), NGF—nerve growth factor, RANTES—regulated upon activation, normal T-cell expressed and secreted (=CCL5), TNFα—tumor necrosis factor α, and TrkA—tropomyosin receptor kinase A.

IFN-γ and TNF-α concentrations were increased after DEE exposure, and higher DEE doses tended to influence the IFN-γ levels to a higher extent than that of TNF-α [29]. This findings are in contrast to the previously described decreased TNF-α and IFN-γ levels found after carbon black exposure; however, as stated before, an equilibrium between a beneficial immune response and exuberant response is crucial [46]. An increased release of TNF-α from human monocytes with a strong additive effect was observed after cigarette smoke extract exposure and RSV infection, and a higher percentage of extreme TNF-α increase was seen in the presence of an RSV + CSE co-exposure, compared to single exposures [53]. 

The vast effects of cigarette smoke exposure were observed in a murine model of repeated RSV infections [38]. A co-exposure provoked an increased in the expression of cytokines (IL-1a, IL-17, IFN-c, KC, IL-13, CXCL9, RANTES, MIF, and GM-CSF), as well as proteases (MMP-2, -8, -12, -13, -16, and cathepsins E, S, W, and Z) [38]. Protein phosphatase 2A (PP2A) and protein tyrosine phosphate 1B (PTP1B) seem to play a significant part in both effects, as they neutralize inflammation and protease expression [38]. This finding was further explored in a study on Ptp1b-deficient mice showing damage in epithelial cell barriers, an enhanced influx of immune cells and cytokine production, and increased apoptosis [39]. 

Interestingly, except for the pathogen-associated molecular pattern (PAMP) triggered by RSV, the damage-associated molecular pattern (DAMP) also seems to be involved in the mechanism [39,93,94]. DAMP consists of an inflammatory response induced by molecules released from infected, damaged, or dead cells; in this study, an increased expression of S100A9 was shown in the lungs of Ptp1b -/- mice [39,95]. Under normal conditions, PTP1B suppresses S100A9 expression during an RSV infection, while an enhanced secretion of S100A9 with resulting inflammation was seen in wild-type mice exposed to cigarette smoke as a result of a desensitized PTP1B activity, as well as in differentiated human bronchial epithelial cells from COPD donors after an RSV infection [39]. S100A9 induces a cytokine release (MCP-1, CXCL10, IL-8, RANTES, G-CSF) from small airway epithelial cells in a TLR4 -dependent manner, enhancing lung damage [39]. In fact, Ptp1b-deficient mice showed an increased RSV-induced immune cell influx, damaged epithelial cell barriers, and an increased apoptosis [39]. The use of anti-S100A9 antibody reduced the immune cell influx into the lungs and perivascular inflammation, as well as apoptosis [39]. 

IL-8 gene and protein expression was also increased in airway epithelial cells after co-exposure to cigarette smoke concentrate (CSC) and RSV infection, together with the augmented monocyte chemoattractant protein-1 (MCP-1) expression [34]. The interferon stimulatory response element (ISRE) site of the IL-8 promoter plays a crucial role in the mechanism; an activation of the interferon regulatory factor-1 and 7 (IRF-1 and IRF-7), which bind to this region, is enhanced in response to a CSC + RSV co-stimulation, thus promoting IL-8 gene transcription [34]. The nuclear factor kappa B (NF-kB) also binds to the IL-8 promoter, synergistically augmenting the IL-8 gene transcription upon a CSC + RSV co-exposure [34]. A CSC enhanced NF-kB–driven IL-8 gene transcription is observed not only after 6 h (like in the case of RSV alone), but also 12 and 24 h post infection [34]. Therefore, an exuberant immune response may be stimulated by two different pathways, and may persist longer than in the case of a single stimuli [34]. 

#### 4.3.2. Airway Hyperresponsiveness (AHR)

An enhanced production of inflammatory cytokines might cause increased airway reactivity. While the RSV itself has the ability to induce the airway hyperresponsiveness (AHR), and a discussion on the causal relationship with the recurrent chronic wheezing is active, an additional effect of air pollution might also be expected [96]. 

The role of increased levels of MCP-1 might be pivotal for AHR. MCP-1, except for its role in the monocyte chemotaxis and lymphocyte differentiation, might be related to an inappropriate activation of a Th2-dependent immune response [83]. The RSV was shown to induce the production of MCP-1 and MIP-1α in the epithelium of the small airways and lungs; MCP-1 and MIP-1α are strong chemo-attractants for eosinophils, which further release RANTES and MIP-1-α; RANTES production is also synergistically increased by a combination of IFN-γ and RSV infection [97]. A plausible explanation for the role of MCP-1 was offered in an experimental model where MCP-1 inhibition resulted in a decreased AHR, and a clinical investigation showed increased levels of MCP-1 in asthmatic patients compared with non-asthmatic patients [83,98]. Of note, RSV increases MCP-1 expression to a higher degree than human rhinovirus, parainfluenza virus, or adenovirus [83,99].

The murine model of carbon black (CB) exposure and an RSV infection showed enhanced AHR in challenge with methacholine, and increased levels of MCP-1 and MIP-1α in CB + RSV-exposed animals, while RANTES was elevated irrespective of carbon black exposure, except for day 14, when it was statistically higher only in CB + RSV mice [47]. MCP-1 levels were elevated on day 4, 5, and 7, but MIP-1α continued to be elevated on day 14 as well [47]. 

Previously, it had been postulated that RSV-induced NGF release causes alterations in the reactivity of sensory nerves and their distribution in the respiratory tract, promoting neurogenic inflammation and hyperreactivity [100]. The study on human bronchial epithelial cells exposed to TiO_2_-NP showed an additive effect on NGF and TrkA (which is a high-affinity receptor for NGF) gene expression with a decreased expression of the low-affinity receptor p75NTR [36]. In an animal model (rats), chronic exposure to nicotine resulted in increased NGF expression, and created a receptor imbalance by favouring the expression of proinflammatory p75NTR, but whereas a significantly higher NGF expression was observed in those animals after an RSV infection, there were no statistical differences in the receptor expression [55]. It also needs to be underscored that a sequence and duration of the host response might play a role, since NGF BALF levels increased in RSV-infected mice from day 14 to 60, while the IFN-γ concentrations were increased on days 7 to 14 [101]. The administration of resveratrol in that model decreased the NGF levels, airway inflammation, and AHR [101].

LIF (leukemia inhibitory factor) is a cytokine (IL-6 class) regulating the inflammatory response and acting in a protective manner against lung injury [102,103,104,105]. Except for tissue protection during pneumonia via the activation of STAT3 (signal transducer and activator of transcription 3) [105], and an unequivocal role in apoptosis (some authors suggest it has proapoptotic properties [106,107], while others find it anti-apoptotic [108,109]), LIF had been previously shown to play a role in airway hyperresponsiveness; its neutralization enhances AHR and airway reactivity to methacholine [102,110]. LIF might be regulated by various factors, including an inhibition by IFN-γ or an enhancement by IL-1β [111,112]. A study on human subjects (BALF from volunteers: never smokers, smokers, and COPD patients) revealed lower LIF protein concentrations in BALF from smokers and COPD patients [51]. Human bronchial epithelial cells obtained from COPD patients BALF showed much lower LIF production upon RSV infection compared to healthy volunteers’ epithelial cells [51]. An animal model also disclosed a lower LIF and its receptor, LIFR, levels during an RSV infection in mice exposed to cigarette smoke [51]. 

Another protective mechanism is related to the cystathionine γ-lyase enzyme (CSE), which generates H2S in the lung [44]. CSE-deficient mice exposed to tobacco smoke for 2 weeks showed an augmented AHR to methacholine challenge, and after an RSV infection, the AHR was increased compared to wild-type mice [44]. CSE-deficiency also resulted in enhanced viral titers and inflammatory cytokines and chemokines levels, including IL-12, G-CSF, TNF-α, MCP-1 (CCL2), MIP-1α (CCL3), MIP-1β (CCL4), and RANTES (CCL5) [44]. 

#### 4.3.3. Histopathological Changes

Histopathological findings may reflect both short- and long-term effects of air pollution and RSV exposure. An RSV pneumonia study by Hashiguchi (mice exposed to TiO_2_-NP) showed an increased lymphocyte infiltration in alveolar septa [41], and a proportion of alveolar septum tissue was used as one of pneumonia severity markers [41]. The study revealed much thicker septum tissue in mice co-exposed to TiO_2_-NP and RSV [41]. In line with those findings, an interstitial inflammation was also noted as a septal thickening in carbon black-exposed mice infected with RSV [46]. Airway wall thickening and a dose-dependency was reported after TiO_2_-NP exposure, and immune cells were accumulated in the peribronchial spaces, which seem to be especially vulnerable regions [52].

The histology of the lungs of mice exposed to DEE showed a dose-related lung inflammation and airway remodelling with a predominance of AM in BALF [29]. Low levels of DEE provoked a peribronchial inflammatory cell infiltration, while high doses resulted in both a peribronchial and peribronchiolar inflammation [29]. Alterations in the airway epithelium structure were induced by DEE regardless of its dose, yet high-level DEE resulted in cell sloughing [29]. Surfactant protein dysregulation was reported and a dose-dependent decrease of surfactant proprotein B (proSP-B) was observed, suggesting alteration in the alveolar function [29]. Moreover, increased metaplasia was noticed in the airway epithelium mucous cells [29].

Histopathological studies in repeated RSV infections found an enhanced influx of macrophages, neutrophils and lymphocytes, located mostly in the perivascular and peribronchial regions with an enhanced airspace enlargement and fibrosis, and a significant apoptosis [38]. Apoptotic response, together with cytokine and protease release, are crucial factors leading to a post-RSV infection airspace enlargement, which results in severe emphysema, as shown by Mebratu [48]. RSV enhances the inflammatory response in cigarette smoke-exposed mice, with an increased mRNA expression of IL-17, IL-1β, IL-12b, IL-18, IL-23a, MCP-1 (Ccl-2), Ccl-7, and MMP-12 [48]. A higher IL-17 expression correlates with higher MMP-12 mRNA expression, which plays a significant role in the formation of emphysema [48,113]; a stimulation of the IL-17 → MCP-1(Ccl-2) → MMP-9 → MMP-12 axis is crucial [48,114]. Cigarette smoke seems to act as a selective adjuvant and enhance Th17 cell differentiation; mice lacking the IL-17 receptor A (IL-17RA) did not develop emphysema after a prolonged cigarette smoke exposure [114].

MMP-12 produced by alveolar macrophages in response to an RSV infection has recently been shown to exacerbate allergic airway inflammation and AHR [115]. MMP-12, one of the matrix metalloproteinases, is an enzyme produced mainly by macrophages; it exerts various biological effects, such as the degradation of elastin fiber or the degradation of type I IFN [116]; the latter one may diminish the antiviral response, increase viral titers, and promote neutrophil accumulation in the lungs [115,117]. Increased MMP-12 levels are related to COPD, and a study by Makino shows that continuous MMP-12 production (in response to cigarette smoke, for example) is necessitated to develop emphysema, while neutrophil infiltration exacerbates airway inflammation [115,118,119,120].

## 5. Discussion

In this scoping review, we identified 25 papers addressing the molecular mechanism of the RSV–air pollution interaction. This review indicates a significant lack of research, taking into account both the frequency of RSV infections and the global air pollution issue. Although the distribution of the studies is very narrow (the vast majority was carried out in the USA), the studies clearly present high scientific quality, however, it should be treated as a call for action for other scientific centres. A more distressing conclusion is that we do not observe an increase in the number of studies in the last years—the number is not growing and the “hockey stick” phenomenon is definitely not observed. The major knowledge gaps may be divided into 3 main areas of interest:-air pollutants: the highest number of studies (although still not enough to identify and confirm the most important pathways in different models) that investigated the interaction between the RSV and cigarette smoke (or its derivatives); the data on the mechanisms of other pollutants is very scarce. While the studies on cigarette smoking and its relationship with the RSV are driven by the issue of COPD exacerbations due to infections (among which the RSV plays an important role), the influence of the other pollutants is hugely under-investigated. As for the six most significant air pollutants (according to the WHO: PM_2.5_, PM_10_, O₃, NO₂, SO₂ and CO) [1], there are only single studies;-the mechanisms facilitating an acute infection or worse disease course: the problem is of great clinical relevance; although the number of clinical studies underlying the relationship between the air pollutants and RSV morbidity is growing, the mechanisms of the interaction remain deeply unknown. We qualified the mechanisms underlying the increased morbidity together with a worse clinical course, since the studies on morbidity are performed mostly in hospital settings, and in fact present a problem of combined increased morbidity and severity, thus, common pathways might also be expected. An explanation of the mechanisms is of special meaning for short-term interventions, especially non-pharmacological ones, which might decrease RSV morbidity during periods of high air pollution;-the mechanisms of an inadequate response to the infection, resulting in a prolonged course of the acute infection, and probably related to long-term sequelae, such as the airway hyperreactivity following an RSV infection; here, a combination of air pollution and RSV infection might be particularly detrimental, and molecular mechanisms need to be well understood in order to take targeted actions, such as pharmacological interventions. In this regard, the scoping review identified some prospective targets for future considerations. Of interest, co-exposure to higher air pollution and RSV infection might play a role in long-term sequelae, and in this regard deserve more attention in clinical settings as well.

This scoping review also has certain limitations. In order to identify the mechanisms as broadly as possible, we included studies focusing on the effects of various air pollutants, while the effects might be pollutant-specific. Moreover, the study included various models (human, but also animal, including various species), and since the effects might be model-specific, we cannot assess with certainty how they translate to human beings; furthermore, important differences between human beings with regards to age groups (infants versus the elderly, for example) might be expected. The number of studies included in the scoping review is small compared to the possible interactions with the number of air pollutants and vast pathophysiological effects; obviously, this might be related to the search bias, but since a systematic approach was used for the search, this problem seems to be related to the low number of published data itself.

## 6. Conclusions

The scarcity of evidence on the molecular bases of the RSV–air pollution interaction presents a serious problem with regards to future actions against the air pollution effects. The accumulated data does not answer the question of how RSV morbidity or disease course are influenced by air contaminants in human beings, and only partially suggests which pathomechanisms are involved. On the other hand, while a growing body of evidence relates the increased number of RSV cases to air pollution, the lack of scientific knowledge on the molecular basics might impede a public discussion on actions against air pollution; we observe the effects without in-depth knowledge of the mechanisms involved. The evidence is also insufficient to suggest any targeted interventions, although some directions have been mentioned. Future studies should also undertake the task to compare air pollutants in terms of the extent of their deleterious effects on the molecular level, as well as focus on different pollutants’ concentrations in order to establish tolerable pollution levels as precisely as possible. In summary, we believe that such important objects as RSV and air pollution deserve and guarantee further studies.

## Figures and Tables

**Figure 1 ijms-23-12704-f001:**
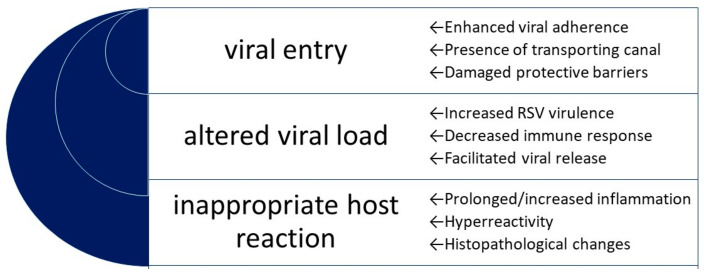
A schematic presentation of the expected directions of air pollution influence on RSV morbidity and disease course; three major concepts are shown in the centre, alongside the corresponding putative pathways on the right.

**Figure 2 ijms-23-12704-f002:**
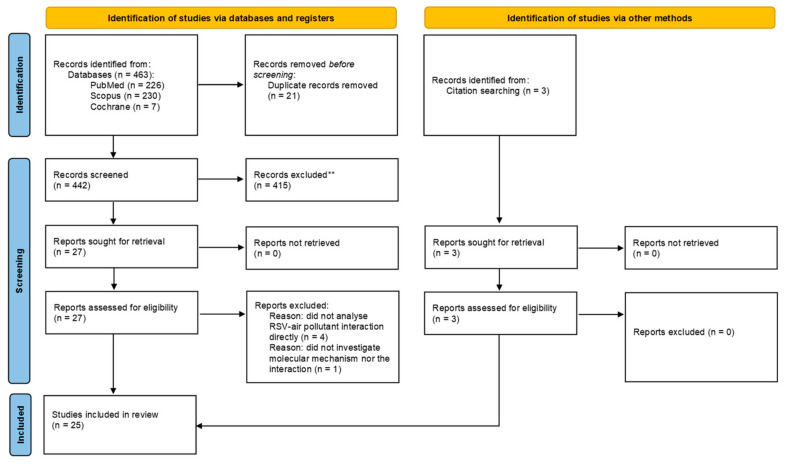
PRISMA flow diagram.

**Figure 3 ijms-23-12704-f003:**
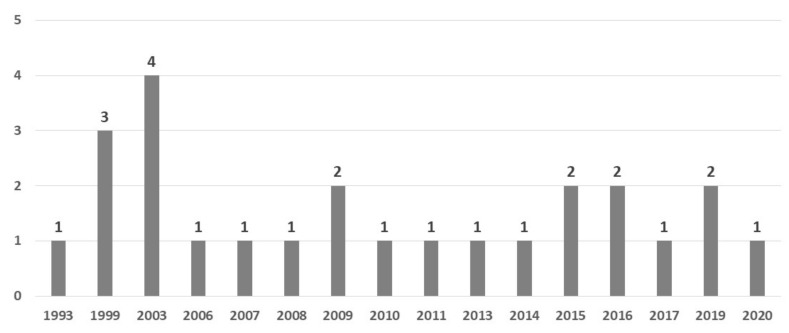
The number of studies included in the scoping review according to the year of publication.

**Figure 4 ijms-23-12704-f004:**
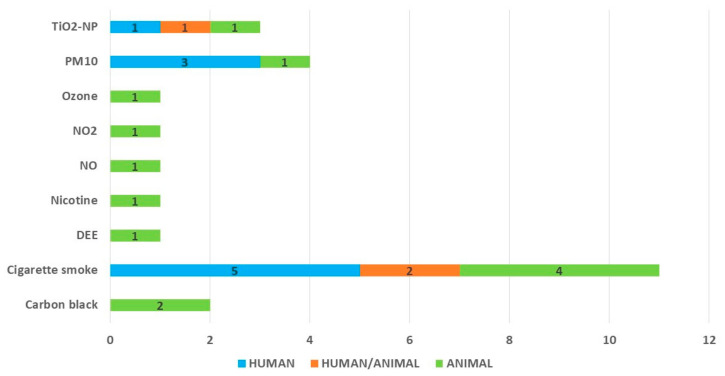
The number of studies investigating the particular air pollutants according to the model used in the study: blue—human model, orange—mixed human and animal model, green—animal model. TiO_2_-NP.

**Figure 5 ijms-23-12704-f005:**
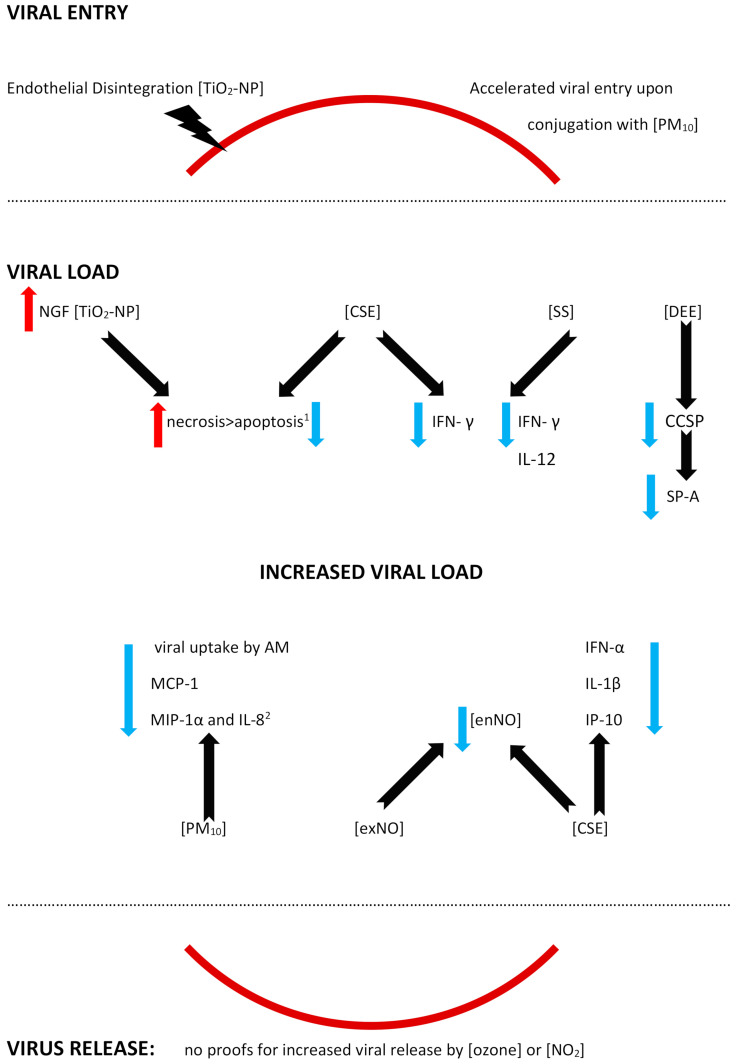
A diagram of the feasible mechanisms facilitating an RSV infection upon an exposure to the different air pollutants. The activity of the air pollutants is shown with the black solid arrow, while the directions of the effects are marked with red (increase) or blue (decrease) arrows; the red solid line symbolizes a division between the extracellular and intracellular space (in the case of a viral entry and a viral release), and the red solid lines are separated from the rest of the figure by black dotted lines, since the remaining effects (shown in the central part of the scheme) are reported in extra- or intracellular spaces. Please note: the figure has only a demonstrative character and is based on the literature found for the purposes of this review. The effects of the air pollutants might vary with regards to the presence/absence of the mechanism, its extent, and its effects. The mechanisms might be influenced by other, not verified or not shown, pathways. The mechanisms might be dose-dependent, time-dependent, exposition sequence-dependent, and model-dependent (i.e., differences between the models used in the studies and human beings might be seen). The figure is simplified, and, for example, the viral load here is a common understanding of the enhanced presence of an RSV load, measured in different ways, and depending on the study (increased expression of the RSV genes and/or proteins, viral titers, reappearance of replication, etc.); for details, see the text and/or refer to the original articles. In addition, contradictory effects of the same air pollutant might be reported, depending on the study model. Abbreviations: CSE—cigarette smoke extract, DEE—diesel engine emissions, NO_2_—nitrogen dioxide, PM_10_—particulate matter smaller than or equal to 10 µm in diameter, SS—s ide-stream cigarette smoke, TiO_2_-NP—titanium-dioxide nanoparticles; AM—alveolar macrophage, CCSP—Clara cell secretory protein, IL-(1β, 8, 12)—Interleukin-(1β, 8, 12), IFN-α—interferon α, IFN-γ—Interferon γ, IP-10- IFN—γ-inducible protein (=CXCL 10), MCP-1—monocyte chemoattractant protein-1 (=CCL2), MIP-1α—macrophage inflammatory protein-1 α (=CCL3), ex/enNO—exogenous/endogenous nitrogen oxide, and SP-A—surfactant protein A. ^1^—increased cell death (probably due to necrosis) and a decreased apoptosis in the case of cigarette smoke extract, and ^2^—a lack of an additive effect, which may be interpreted as a relative decrease.

**Figure 6 ijms-23-12704-f006:**
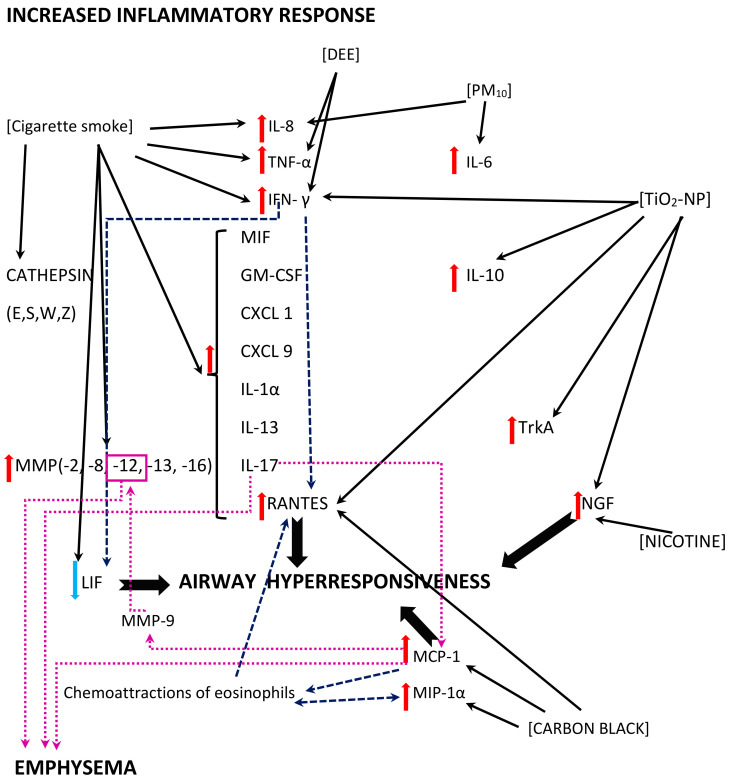
A diagram of the exuberant inflammatory response and its effects; multidirectional interactions between the air pollutants (shown in square brackets) and inflammatory mediators, including cytokines and chemokines are shown. The black solid line arrows reflect direct effects of the air pollutants revealed in the studies; the dashed navy blue line arrows reflect additional effects of the inflammatory mediators or immune cells contributing to the effects; the dotted magenta line arrows reflect the mechanisms related to the pathomechanism of emphysema.

**Table 1 ijms-23-12704-t001:** Eligibility criteria for the study inclusion.

Inclusion Criteria	Exclusion Criteria
Original study	Review article
Study performed either on cellular, animal, or human models	Analysis on the molecular mechanism(s) of air pollution only
Analysis of the interaction between air pollutants and RSV	Analysis on the molecular mechanism(s) of RSV only
Investigation on molecular mechanism(s), i.e., altered immune system response, including cytokine/chemokine production and/or release, receptor stimulation, gene expression, histological study	Lack of a potential pathomechanism analysis
	Only prenatal exposure to RSV or air pollution
	Lack of a possibility to obtain a full-text article

**Table 2 ijms-23-12704-t002:** Studies included in the scoping review.

Author	Air Pollutant	Material/Subjects	Country
Becker and Soukup, 1999 [32]	NO_2_ (different concentrations: 0.5, 1.0, and 1.5 ppm)	Human bronchial epithelial cell line BEAS-2B	USA
Becker and Soukup, 1999 “Exposure to urban air particulates alters…” [33]	PM_10_ (EHC-93)	human bronchial epithelial cell line BEAS-2B subclone S6 and human alveolar macrophage from BAL from volunteers	USA
Castro, 2008 [34]	Cigarette smoke condensate	A549, human alveolar type II–like epithelial cells, and 293, a human embryonic kidney epithelial cell line	USA
Castro, 2011 [35]	cigarette smoke extract	Human plasmacytoid dendritic cells	USA
Chakraborty, 2017 [36]	TiO_2_-NP	Human primary bronchial epithelial cells)	USA
Cruz-Sanchez, 2013 [37]	Mimics of ambient particulate matter (PM_10_)	human epithelial-2 (HEp-2) cells 1HAEo-cells- normal human airway epithelial cells transformed with Simian virus 40	Canada
Foronjy, 2014 [38]	Cigarette smoke	C57BL/6J mice (repeated RSV exposition- 6 times)	USA
Foronjy, 2016 [39]	Cigarette smoke	lung BALF from age-matched healthy control subjects, smokers, and subjects with COPD; Ptp1b (Ptpn1 gene) knockout (-/-) mice; human primary small airway epithelial and mouse bone marrow derived macrophages	USA
Groskreutz, 2009 [40]	Cigarette smoke extract	Primary human tracheobronchial epithelial	USA
Harrod, 2003 [29]	Diesel engine emissions	C57Bl/6 mice	USA
Hashiguchi [41]	TiO_2_-NP	BALB/c mice	Japan
Hirota, 2015 [42]	PM_10_ (EHC93)	human airway epithelial cell line (HBEC-6KT)	Canada
Hobson and Everard, 2007 [43]	Nitric oxide	human monocyte-derived dendritic cells (DCs)	UK
Ivanciuc 2019 [44]	side-stream tobacco smoke	cystathionine γ-lyase enzyme (CSE)- deficient and wild-type mice	USA
Kaan and Hegele, 2003 [45]	PM_10_ (EHC-93)	Guinea pig alveolar macrophages	Canada
Lambert, 2003 “Effect of Preexposure to Ultrafine Carbon Black…” [46]	Preexposure to ultrafine carbon black	BALB/c mice	USA
Lambert, 2003 “Ultrafine Carbon Black Particles Enhance…” [47]	Ultrafine carbon black after RSV infection	BALB/c mice	USA
Mebratu, 2016 [48]	Cigarette smoke	C57BL/6 mice	USA
Modestou, 2010 [49]	Cigarette smoke extract	Human trachea and bronchial samples Primary human tracheobronchial epithelial cells	USA
Phaybouth, 2006 [50]	side-stream cigarette smoke	Newborn BALB/cmice (RSV infection twice)	USA
Poon, 2019 [51]	Cigarette smoke-mice COPD patients Smokers	Mice exposed to cigarette smoke; BALF from healthy never smokers, smokers, and COPD patients; Human bronchial epithelial (HBE)	USA
Smallcombe, 2020 [52]	Titanium dioxide nanoparticles	Immortalized human bronchial epithelial cells; C57BL/6 mice	USA
Raza, 1999 [53]	water-soluble cigarette smoke extract (CSE), nicotine, cotinine	monocytes of the blood from donors	UK
Soukup, Koren, and Becker, 1993 [54]	ozone	Human alveolar macrophages	USA
Urrego, 2009 [55]	Nicotine exposure	Rats	USA

**Table 3 ijms-23-12704-t003:** A risk of bias assessment (critical appraisal) with the use of Joanna Briggs Institute Checklist for Quasi-Experimental Studies (non-randomized experimental studies); for detailed domains, (numbered D1–D9) please refer to Appendix A. Abbreviations: Y: yes, N: no, U: unclear. Model used in the study—H: human, A: animal, H/A: mixed human and animal. Reference of the guidelines followed and mentioned by particular papers: ^A^ Institutional Animal Care and Use Committee (IACUC) guidelines, ^B^ Guide for the Care and Use of Laboratory Animals of the National Institutes of Health and Institutional Animal Care and Use Committee, ^C^ unspecified, ^D^ Guideline of the Kyushu University of Health and Welfare, ^E^ Guide for the Care and Use of Laboratory Animals of the National Institutes of Health, ^F^ Canadian Council on Animal Care (CCAC). 1980. Guide to the Care and Use of Experimental Animals, ^G^ Association for Assessment and Accreditation of Laboratory Animal Care-approved guidelines and protocols, ^H^ Guide for the Care and Use of Laboratory Animal of the National Institutes of Health, ^I^ National Institutes of Health Guide for the Care and Use of Laboratory Animals.

		JBI Critical Appraisal Checklist For Quasi-Experimental Studies
Articles	Model	D1	D2	D3	D4	D5	D6	D7	D8	D9
Becker and Soukup, 1999 [32]	H									
Becker and Soukup, 1999 “Exposure to urban air particulates alters…” [33]	H									
Castro, 2008 [34]	H									
Castro, 2011 [35]	H					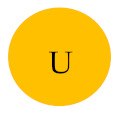				
Chakraborty, 2017 [36]	H									
Cruz-Sanchez, 2013 [37]	H									
Foronjy 2014 [38] ^A^	A									
Foronjy, 2016 [39] ^B^	H/A									
Groskreutz, 2009 [40]	H					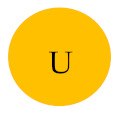				
Harrod, 2003 [29] ^C^	A									
Hashiguchi [41] ^D^	A									
Hirota, 2015 [42]	H									
Hobson and Everard, 2007 [43]	H									
Ivanciuc, 2019 [44] ^E^	A									
Kaan and Hegele, 2003 [45] ^F^	A									
Lambert, 2003 “Effect of Preexposure…” [46] ^C^	A					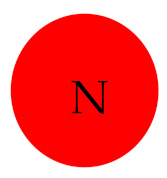				
Lambert, 2003 “Ultrafine Carbon Black …” [47] ^C^	A					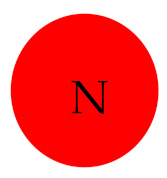				
Mebratu, 2016 [48] ^C^	A					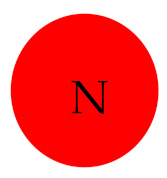				
Modestou, 2010 [49]	H									
Phaybouth, 2006 [50] ^G^	A									
Poon, 2019 [51] ^H^	H/A									
Smallcombe, 2020 [52] ^I^	H/A					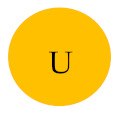				
Raza, 1999 [53]	H					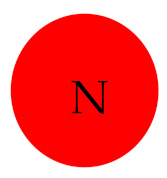				
Soukup, Koren, and Becker, 1993 [54]	H									
Urrego, 2009 [55] ^C^	A									

## Data Availability

Data supporting reported results can be obtained on request from the authors.

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
