# Peer review of "Molecular Mechanisms of RSV and Air Pollution Interaction: A Scoping Review"

_ijms, 2022, doi:10.3390/ijms232012704_

Round 1
Reviewer 1 Report
This review is well written in terms of the logic flow and literature examples. The authors firstly proposed the major issue that they wanted to solve, and gathered extensive examples to set the base. The discussion on the molecular mechanisms in the air-pollutant interaction was adequate and clear. Some limitations were also mentioned. However, one concern is the table 2. I was wondering if the authors can rotate the table by 90 degrees as this won't drag the table too long. Otherwise the writing is acceptable.
Author Response
Dear Reviewer,
Thank you very much for your kind review and the suggestion regarding the table. In fact, we tried to rotate it or shorten it, and it was our concern although we did our best to make it as short as possible. I am afraid that rotating the table did not do the job (it is also too long), thus, I prepared a shortened version of the table (without the results, just citing the studies and their most important aspects) and according to the suggestions of the second reviewer I put the full table with the results in the supplementary materials.
Best regards,
August Wrotek
Reviewer 2 Report
Wrotek and Jackowska publish a thorough review of papers that examine the impact of various pollutants and cigarette smoke on RSV infection and its outcome. There is scarce literature on this topic, and insufficient to delinate clear mechanisms or identify interventions. Yet it is an important area in RSV research and one that can contribute to our understanding and resolution of consequences from RSV infection. Objective criteria are described that were used to select qualifying papers. Impacts of pollutants on entry, replication (load), and antiviral defense are described. The study contains a lot of interactions between immune cells and mediators. Many of these have been investigated and are very complex even without factoring in pollutants, and there is not much awareness or appreciation of how pollution and cigarette smoke influence the outcome of all these interactions. The paper helps the reader appreciate pollution impacts by describing numerous, perhaps somewhat lesser-known but nevertheless interesting, pollution impacts on RSV infection outcome, including TLRs, cytokines, IRFs, PAMPS and DAMPS, and shows in an overview in Fig. 6 which of these impacts are direct versus indirect. It is a rather lengthy read but altogether a brave effort to pull together and compare and discuss the available literature on this topic. In the discussion, the authors describe limitations of the study and the main knowledge gaps that need to be addressed to get to a better understanding of pollution impact. This work should be of interest to many in the RSV field that study inflammation, airway hyper responsiveness, and related pathology.
Comments
There are only minor comments.
- In the 'results' section of the abstract, the link between pollution and RSV disease is not well presented.
- There are some minor language issues here and there, such as 'consciousness' on page 2 should be 'awareness', though overall it is well written and understandable.
- The paper does a good job of bringing together the available literature on this topic in a single read, and providing insights and interpretation.
- The table that runs from page 7 - 19 is very long. It might be desirable to use a different format that compacts the data or include as supplementary data instead.
Author Response
Dear Reviewer,
thank You very much for your review of our paper and all the hints to improve this manuscript! We would like to answer point-by-point the suggestions raised in Your review:
- In the 'results' section of the abstract, the link between pollution and RSV disease is not well presented.
In fact it was not, we changed the abstract, I hope it is clearer now.
- There are some minor language issues here and there, such as 'consciousness' on page 2 should be 'awareness', though overall it is well written and understandable.
Please excuse me. I changed it, those are definitely two separate terms, and a mistake that should not be made.
- The paper does a good job of bringing together the available literature on this topic in a single read, and providing insights and interpretation.
Thank You very much, we did our best.
- The table that runs from page 7 - 19 is very long. It might be desirable to use a different format that compacts the data or include as supplementary data instead.
It was our concern from the very beginning, thus we changed it into shorter table just citing the included studies and presented the full table in the supplementary materials.
Best regards,
August Wrotek